# Monitoring of the Training Load and Well-Being of Elite Rhythmic Gymnastics Athletes in 25 Weeks: A Comparison between Starters and Reserves

**DOI:** 10.3390/sports10120192

**Published:** 2022-11-28

**Authors:** Iohanna Fernandes, João H. Gomes, Levy de Oliveira, Marcos Almeida, João G. Claudino, Camila Resende, Dermival R. Neto, Mónica Hontoria Galán, Paulo Márcio P. Oliveira, Felipe J. Aidar, Renata Mendes, Marzo E. Da Silva-Grigoletto

**Affiliations:** 1Functional Training Group, Post Graduate Program in Physical Education, Department of Physical Education, Federal University of Sergipe, São Cristóvão 49100-000, Brazil; 2Department of Physical Education, Federal University of Sergipe, São Cristóvão 49100-000, Brazil; 3L’Esporte—Exercise and Sport Performance Research Group, Post Graduate Program in Physical Education, Department of Physical Education, Federal University of Sergipe, São Cristóvão 49100-000, Brazil; 4Group of Research, Innovation and Technology Applied to Sport (GSporTech), Multi-User Laboratory of the Department of Physical Education (MultiLab of the DPE), Department of Physical Education, Center for Health Sciences, Federal University of Piauí, Teresina 64000-850, Brazil; 5Brazilian Gymnastics Federation, Aracaju 81610-020, Brazil; 6Faculty of Physical Activity and Sports Sciences (INEF—Sports Department), Polytechnic University of Madrid, 28040 Madrid, Spain; 7Department of Physical Therapy, Federal University of Sergipe, Lagarto 49100-000, Brazil; 8Study Group on Nutrition Applied to Exercise (GENAE), Department of Nutrition, Federal University of Sergipe, São Cristóvão 49100-000, Brazil

**Keywords:** session-RPE, internal training load, sports performance, training and testing, readiness

## Abstract

The objective of this study was to monitor the training loads (TL) and well-being of elite rhythmic gymnastics (RG) athletes, as well as compare these variables between starters and reserve gymnasts during 25 weeks of training. Ten athletes from the Brazilian national RG team (17.4 ± 1.1 y of age) were monitored during the general preparatory period (GPP), specific preparatory period (SPP), and pre-competitive period (PCP). The internal TL was quantified with the use of sessional ratings of perceived exertion (sRPE). We assessed well-being daily with a well-being scale. The TL, duration, monotony, and strain were calculated weekly. We found that the internal TL and session durations were 9242 ± 2511 AU and 2014 ± 450 min, respectively. The internal TL, strain, and monotony were greater in the PCP than in the GPP and SPP for starters. In the SPP, there were statistical differences in internal TL (*p* = 0.036) and strain (*p* = 0.027) between starters and reserves. In the PCP, there were also statistical differences between starters vs. reserves athletes regarding internal TL (*p* = 0.027) and strain (*p* = 0.05). There was no statistically significant difference in well-being between the periods assessed. In conclusion, RG athletes display a higher TL magnitude during the PCP, whereas only reporting non-significant minor variations in well-being. In addition, there is a discrepancy in the TL between starters and reserves.

## 1. Introduction

Rhythmic gymnastics (RG) is a sport that can be practiced individually or in a group, where athletes are subjectively evaluated through the code of points [1,2,3,4]. As the sport evolved, there was an increase in the weekly training volume, and consequently, the training load (TL) may also have been affected [5]. Thus, RG athletes’ perception of recovery may be influenced by the high TL that they are exposed to [6], and this high TL may be related to injuries resulting from repeated movements or excessive use [7,8,9]. Therefore, investigating the relationship between training load, performance, and injury risk would provide a stronger case for monitoring TL in this population [10].

A recent study [11] conducted with coaches, medical staff, and gymnasts from 25 countries showed that external TL (i.e., the physical work performed during training) in RG was often calculated by the total number of repetitions and training duration. The authors concluded that the internal TL (i.e., psychophysiological response) was usually estimated through the coach’s perception of an athlete’s effort [11]. Moreover, half of the coaches perceived poor TL management in the modality [10], highlighting that further investigations on tools for better distribution of TL in the modality should be carried out.

Debien et al. [12] analyzed an entire season of elite RG athletes and found high training loads (TLs) (10,381 ± 4894 Arbitrary Units (AU)), especially during the competitive period (13,391 ± 3392 AU). The higher TLs experienced in the competitive period can affect an athlete’s physical and psychological well-being, potentially leading to higher overall fatigue and increased injury risk [13]. As a result, it may be warranted to monitor the well-being of RG athletes as indicators of overreaching that are associated with mood, sleep, and stress disorders can easily be quantified by these scales [13]. This monitoring may assist coaches in their decision-making processes to minimize the risk of injuries and improve performance. To the best of our knowledge, the general well-being of young gymnasts was evaluated in only one study during six weeks of training with different TL, and the authors found no differences in the athletes’ general well-being over this period [14].

Another potential consideration is that there may be differences in the programmed TL and wellness scores between starters and reserves. According to Debien et al. [15], during the competitive period before the 2016 Olympic Games, four starters out of six gymnasts from a team suffered overuse injuries over 126 days. This finding points to a complex scenario as the substitution of starter athletes tends to change the collective performance, with each athlete having a specific role in the group’s performance [16]. Furthermore, according to Debien et al. [15], of the six athletes monitored, the two reserve athletes in most competitions in the analyzed period, were the starters in the 2016 Olympic Games. This information may demonstrate the importance of potential differences in the TL and ratings of well-being between starters and reserves. The potential differences in the TL and well-being ratings experienced by starters and reserves may impact the whole team’s performance, as reserve athletes may not be submitted to the same training load.

Given the above, it is important to investigate the monitoring and distribution of TL in RG and elucidating how TL affects the well-being of gymnasts in starter vs. reserve groups. This information would be helpful for coaches and sports scientists aiming to improve athlete performance and reduce injury risk. Therefore, we aimed to examine the TL and general well-being of elite RG athletes during 25 weeks of training and compare these results in starter versus reserve gymnasts.

## 2. Materials and Methods

### 2.1. Subject

Five starting athletes (17.8 ± 0.8 y; 1.60 ± 0.01 m; 49.6 ± 3.4 kg) and five reserve athletes (17.4 ± 0.5 y; 1.60 ± 0.05 m; 50.6 ± 2.9 kg) from the Brazilian national RG team with 9.9 ± 2.4 y of experience in the sport participated in this study. These athletes took part in the preparatory period for the 2020 Olympic Games. Athletes who could perform the training proposed by the technical committee were included. Athletes who completed <75% of weekly sessions, whether due to injuries or other reasons, were excluded from the statistical analysis, besides those who were not part of the team until the end of data collection, independent of the reasons. All athletes read and signed an informed consent document or assent was given by their legal guardian (<18 y) before participating in the current study. This study was conducted following the Declaration of Helsinki and approved by the Ethics Committee for Human Research of the Federal University of Sergipe (report no.: 4.571.105).

### 2.2. Study Design

During the 2020 training season, data were collected for 25 weeks that were divided into three distinct periods, based on athlete periodized training plans, without the interference of the researchers. Seven weeks were allocated to the general preparatory period (GPP), whereas the specific preparatory period (SPP) involved 12 weeks of training, and the pre-competition Period (PCP) was composed of five weeks of training. During the 13th and 14th weeks, the gymnasts trained only one session a day. Furthermore, at the end of the 25th week (last analyzed week), some gymnasts tested positive for COVID-19, interrupting the study.

### 2.3. Data Collection Procedure

Before the first training session, anthropometric data (body mass and height) were collected from each gymnast using an anthropometric scale that contained a stadiometer (Líder^®^, P150C, São Paulo, Brazil). On all training days during the study, the athletes answered an online well-being form in the morning, and after the first and second training sessions, they rated each session’s perceived exertion (RPE).

### 2.4. Training Load

The duration and frequency of each training session were recorded, and the weekly training duration in minutes was calculated as the sum of all weekly sessions performed. The internal TL was obtained by calculating the sRPE (RPE x duration in minutes) [17]. Thirty minutes after each training session, all athletes answered the following question, “How was your training session?” on an online form, indicating on a scale with values from 0 to 10, where 0 represented “no effort” and 10 represented “maximum effort”. For RPE weekly mean classifications, we adopted: ≥7 = high, >4 to <7 = moderate, and ≤4 = low [18].

Daily internal TL was calculated by summing session (morning and afternoon) training loads, and the weekly internal TL was the summation of each training load over the week. Internal TL was classified according to the range of minimum and maximum values found over the 25 weeks, considering the following thresholds for maximum values: <25% = low, 25–50% = moderate-low, 50–75% = moderate-high, and >75% = high [19]. To obtain monotony, the average weekly TL was divided by the standard deviation of the weekly TL. The strain was calculated by multiplying the monotony by the weekly TL.

### 2.5. Well-Being Perception

A 5-point Likert scale was used to measure well-being [20]. The gymnasts provided ratings for the following domains: general muscle soreness (1—Very sore, 2—Increase in soreness/asleep, 3—Normal, 4—Feeling good, 5—Feeling great), fatigue (1—Always tired, 2—More tired than normal, 3—Normal, 4—Fresh, 5—Very fresh), sleep quality (1—Insomnia, 2—Restless sleep, 3—Difficulty falling asleep, 4—Good, 5—Very restful), stress level (1—Highly stressed, 2—Feeling stressed, 3—Normal, 4—Relaxed, 5—Very relaxed), and mood (1—Highly annoyed/irritable/down, 2—Snappiness at teammates, family, and co-workers, 3—Less interested in others and activities than usual, 4—A generally good mood, 5—Very positive mood). General well-being was calculated by the sum of all domains [14,20].

### 2.6. Statistical Analysis

Data were expressed as mean, maximum values, and standard deviation. Data normality was assessed by the Shapiro–Wilk test and the homogeneity of variances using Levene’s test. The values of internal TL, strain, monotony, training duration, RPE, fatigue, sleep quality, general muscle soreness, stress level, mood, and general well-being of the groups (starters vs. reserves) were compared in the three periods (GPP vs. SPP vs. PCP) through a mixed-design analysis of variance (ANOVA), followed by Bonferroni post hoc tests. Cohen’s effect size (ES), d, was analyzed to determine the magnitude of the effect. Ratings were interpreted as: <0.2 trivial effect; 0.2–0.49 small effect; 0.50–0.8 moderate effect; and >0.8 large effect [21]. The percentage of change (Δ%) was calculated, and the level of statistical significance adopted was *p* ≤ 0.05. Statistical analyses were performed using the program R version 4.1.2.

## 3. Results

### 3.1. General Description

A total of 225 training sessions were performed, with an average of 9.0 ± 1.7 sessions per week. The training sessions lasted between 4 and 5 h. The maximum value of internal TL reached about 30% above average throughout the period, and athletes generally perceived the level of effort as moderate, even when the session was at its highest intensity. Interestingly, the week with peak internal TL did not coincide with the week of peak RPE (Table 1). The longitudinal analysis of these variables, stratified by preparation weeks and periods (GPP, SPP, and PCP), is shown in Figure 1.

Throughout the three periods, 19 (76%) weeks were classified with moderate RPE training and 6 (24%) with low RPE, with no weeks classified as high effort. In addition, 48% of the internal TL were classified as high (≥9850 AU), 40% as moderate-high (≥6566 to <9850 AU), 8% as moderate-low (≥3282 to <6566) AU, and 4% as low (<3282 AU).

Regarding general well-being, fatigue and stress were classified as “normal”, sleep was classified as “good”, mood was described as “generally good mood”, and general muscle pain was classified as “increased muscle pain”. Thus, the general well-being of all athletes over the weeks was classified as “normal”, as suggested by the classification of Antualpa et al. [14] (Figure 2).

### 3.2. Comparisons between Starters and Reserves

There was a statistical difference between the internal TL in the SPP (*p* = 0.03; ES = −1.60; Δ% = −22.4) and PCP (*p* = 0.02; ES = −1.71; Δ% = −33.9) periods between starters and reserves, with starters displaying higher internal TL. Starters displayed a significant increase in internal TL during the SPP (*p* = 0.01; ES = 1.69; Δ% = 23.6) and in the PCP (*p* = 0.02; ES = 1.79; Δ% = 45.4) compared with the GPP (Figure 3A).

Regarding monotony and training duration during the three periods, no statistical differences were found between the groups [F(1,2;10) = 0.2; *p* = 0.7 and F(1.1;8.9) = 0.1; *p* = 0.7, respectively]. However, the starters had higher values for strain during the SPP and PCP than the reserves (*p* = 0.04; ES = −1.52; Δ% = −23.3 and *p* = 0.05; ES = −1.45; Δ% = −29.1) (Figure 3C). When comparing periods, the starter athletes had a lower value of monotony during the GPP compared with the PCP (*p* = 0.04; ES = 2.21; Δ% = 14.1) and reserves in the SPP period compared with the PCP (*p* = 0.04; ES = 1.76, Δ% = 20.9). The training duration for reserve athletes was longer in GPP compared with SPP (*p* = 0.04; ES = −1.85; Δ% = −6.1) (Figure 3B,E). Regarding strain, the GPP of the starter group showed lower values compared with SPP (*p* = 0.01; ES = 1.54; Δ% = 26.7) and PCP (*p* < 0.01; ES = 2.20; Δ% = 68.7).

Regarding the RPE, during the GPP, the starters obtained lower values compared with the SPP (*p* < 0.05; ES = 2.43, Δ% = 26.3) and PCP (*p* = 0.01; ES = 1.91; Δ% = 47.4). Differences were found in the comparisons between groups (*p* = 0.03), with higher values for the starter athletes only in the PCP (Figure 3D). Finally, there was no difference (intra- or inter-group) in general well-being and their domains in the starter and reserve athletes during the periods (*p* > 0.05).

## 4. Discussion

This study aimed to quantify the TL and general well-being of high-performance RG athletes and compare these variables in starter and reserve gymnasts during 25 weeks of training. Our main findings were that 88% of the analyzed weeks had moderate-high to high TL, based on the session-RPE. In 60% of the weeks, athletes were classified with a “normal”/“good” general well-being, according to summed questionnaire domains. Comparisons between groups demonstrated differences in internal TL, strain, and RPE between starters vs. reserves, mainly in the periods closer to the competition.

### 4.1. General Description

The long duration can explain the high internal TL values and frequency of training, commonly seen in RG [9,22], as there were no weeks classified as having a high RPE. Nevertheless, in our research, the internal TL values were lower than those found in another study in RG, where the peak week had a 37.5% higher value [11]. These differences may be explained by the different periods of the season. However, we found higher TL compared with other sports [23,24].

We used the classification of TL, suggested by Miloski et al. [19], and these will vary according to each context. In addition, the scoring code changes in RG, with the requirements modified at each cycle [1,3,4]. Thus, it is essential to understand how these changes influence the characteristics of the competition routine and distributions of TL during each Olympic Cycle.

Using the well-being questionnaire in RG may be necessary due to the sport’s technical predominance. Selmi et al. [25] pointed out several relationships between the technical actions of athletes in soccer and well-being indices (fatigue, general muscle pain, and general well-being). Higher values of fatigue and general muscle soreness were associated with fewer correct passes and correct intercepts, indicating that these domains cause disturbances to the athlete’s technical performance. Variations in well-being domains were not found in the present study, despite the high TL. A possible explanation for these findings may be because RG athletes start training early in life, due to the characteristics of the modality, and consequently have experienced high TL since childhood [9]. Thus, these athletes may be so used to the perceptions of constant TLs that they could have fewer perceptions of changes in general well-being indicators.

### 4.2. Comparison between Starters and Reserves

In some sports (e.g., volleyball and soccer), starters display higher training loads compared with reserves [26,27]. In this study, the starters displayed higher values of internal TL, strain, and RPE than the reserve group in the SPP and PCP periods. These higher values may be related to differences in external TL (not measured in this study), such as significantly greater repetition of routines by the starter group, which may have led to higher RPE, internal TL, and strain [9].

Understanding the internal TL of starters and reserves is paramount for sports. A previous study conducted with soccer athletes investigated physiological and performance responses during a season, and it was found that only starter athletes had decreased performance (jump height and sprint speed) at the end of the season compared with the pre-season, which could put them at risk of non-functional overreaching [28]. This monitoring becomes important in RG because it will be possible to know the responses of each athlete to the TL received. In this sense, TL and general well-being monitoring may be necessary for RG, as this practice may allow coaches to individualize training and make decisions to improve performance [10] and minimize injury risk, mainly caused by overuse [7,8,9].

We did not identify whether the starters were overloaded due to a lack of tests measuring physical performance, but when we analyzed the measures of general well-being in this period for both groups, no differences were found. We emphasize the need for both groups to present similar conditions (e.g., physical, technical, and psychological state), as replacing an athlete from the starter group may cause a disturbance in the system (group) of the team, causing instability [29]; having athletes with similar conditions may imply a shorter reorganization time.

The starters showed differences between GPP vs. SPP and PCP for the internal TL, considering intragroup comparisons in all periods. These findings corroborate other studies with different categories of gymnastics that showed maintenance or increase in internal TL close to competition [12,30,31]. During this period, training tends to focus on technical routines, which are characterized as the most demanding and intense training in RG [9]. Our findings showed no difference in training duration between the periods for the starters; however, there was an increase in RPE in these periods, which may help support this statement.

Although the duration of the sessions did not undergo sudden changes, large volumes of training were found, corroborating another research investigating RG [12,30]. We speculate that the training duration in gymnastics greatly influences the cultural aspect related to the modality [12]. According to these authors, coaches perform many interventions to give feedback to gymnasts, and many repetitions of exercises are performed to achieve good technique, making the sessions longer.

Other indicators were also relevant for TL management. Our findings showed differences in strain for starters during GPP vs. SPP and PCP and monotony in GPP vs. PCP, as well as the monotony of SPP vs. PCP for the reserve group. The strain values in RG were higher than in other team sports reported in the literature [6]. These values seem to be mainly explained by the high internal TL, as the strain is the product of weekly TL vs. monotony [17], and the values found in monotony were within the normal range [17].

Despite this, we found higher monotony during the PCP for both groups, corroborating previously mentioned results [6]. This period represents the moment when there is a greater emphasis on executing competition routines with minimum errors, approaching the most accurate possible conditions of competition that gymnasts will face. We believe that this consistent execution of routines [6] could cause a decrease in the variability of training stimuli. Therefore, we emphasize the need for more significant training variation in these competitive moments to minimize harmful risks to performance.

This study also had some limitations. Among these, the TL was analyzed only through the internal TL, without specific information regarding the external TL. However, given the complexity of counting the elements that make up the external TL of this modality (repetitions of isolated exercises, training density, and number of sets), it would be almost impossible to reliably measure. Another relevant point is that we did not have previous studies evaluating this scale’s sensitivity in RG. However, this tool has been used in other studies in the modality and is widely used by professionals in a practical context. In addition, the number of athletes that composed the sample may offer a statistical limitation. However, this was the entire population of the national elite team of gymnasts, with no possibility of having a larger sample. Our study was also limited to PCP without information from the competitive period due to the athletes infected by COVID-19.

There are several practical applications associated with this study: (a) Considering the implementation of strategies, such as additional training for reserve gymnasts associated with the primary training program, can be beneficial for the improvement and maintenance of physical conditioning, especially in moments closer to the competition; (b) During the competitive period, coaches should plan more significant variation in the TL, avoiding the high monotony in this period that can harm performance and increase the risk of injury; (c) As initial strategies, the use of subjective measures, such as RPE and well-being, should be considered to assist in TL monitoring.

## 5. Conclusions

We conclude that for this RG team, training loads were considered high. In addition, these high training loads seemed to be mainly due to the large volume and frequency of training sessions, in which higher loads were found in periods closer to competition. The general well-being of the gymnasts showed slight variation over the periods. In addition, there was a discrepancy between the training loads for starter and reserve athletes, especially in the specific preparatory and pre-competitive periods.

## Figures and Tables

**Figure 1 sports-10-00192-f001:**
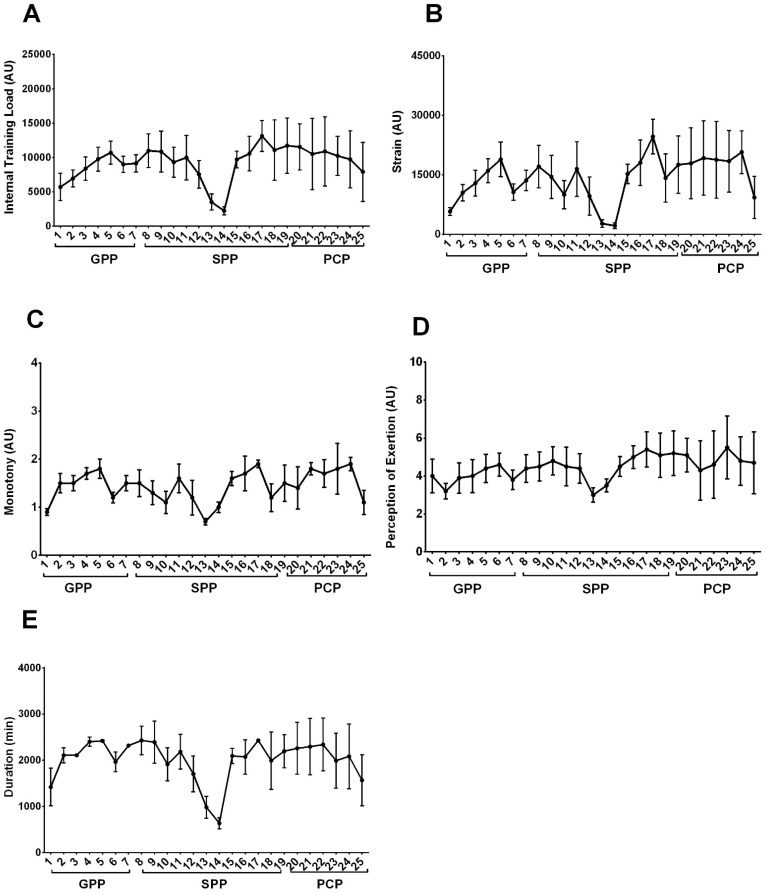
Mean values and standard deviations of internal weekly training loads (**A**), training strain (**B**), training monotony (**C**), perception of weekly effort (**D**), and duration of weekly training (**E**) for 25 weeks. Note. (AU = arbitrary units; GPP = General preparatory period; SPP = Specific preparatory period; PCP = Pre-competitive period).

**Figure 2 sports-10-00192-f002:**
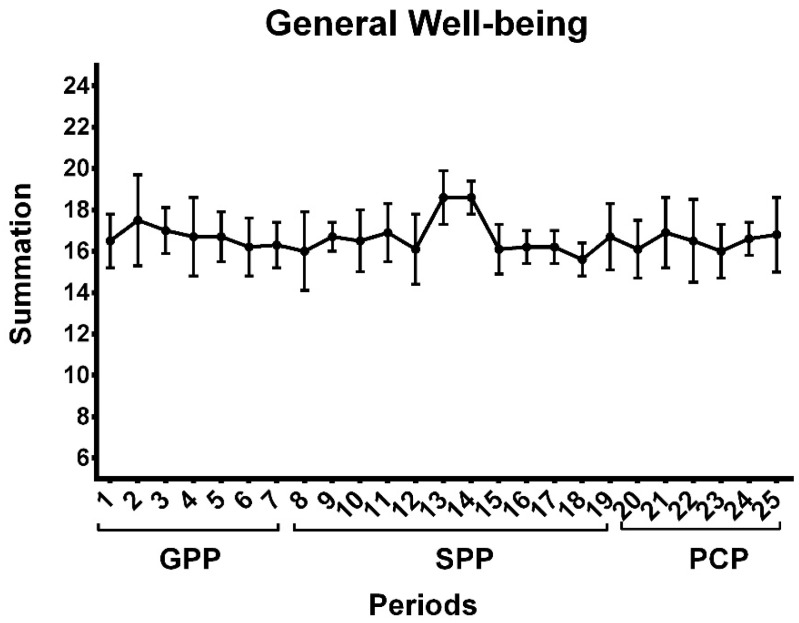
Mean values and standard deviation of the general well-being of all athletes for 25 weeks. Note. (GPP = General preparatory period; SPP = Specific preparatory period; PCP = Pre-competitive period).

**Figure 3 sports-10-00192-f003:**
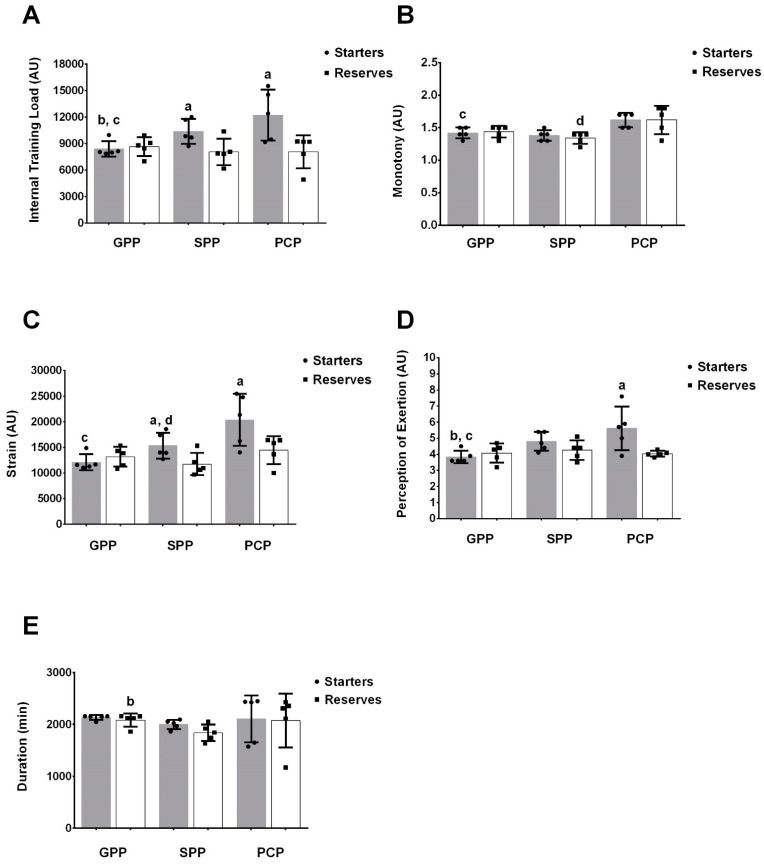
Comparisons between starter and reserve athletes during periods of internal weekly training load (**A**), training monotony (**B**), training strain (**C**), perception of weekly effort (**D**), and duration of weekly training (**E**). Notes: a, Statistically significant difference between starters and reserves; b, Significant difference between GPP vs. SPP; c, Significant difference between GPP vs. PCP, d, Significant difference between SPP vs. PCP (AU = Arbitrary Units, GPP = General preparatory period, SPP = Specific preparatory period, PCP = Pre-competitive period).

**Table 1 sports-10-00192-t001:** Mean, maximum, and peak week values of TL variables for 25 weeks.

Variable	Mean ± SD	Maximum ± SD	Peak Week
Internal TL (AU)	9242 ± 2511	13,133 ± 2259	17
RPE (AU)	4.4 ±0.6	5.5 ± 1.7	23
Monotony	1.4 ± 0.3	1.9 ± 0.08	17
Strain (AU)	14,213.7 ± 5533	24,670.2 ± 4387.2	17
Weekly Duration (min)	2014 ± 450	2430.0 ± 0.0	17

Note. (AU = Arbitrary units, Internal TL = Internal training load, RPE = Rate of perceived exertion).

## Data Availability

Research data are not shared due to privacy and ethical restrictions.

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
