# Peer review of "Monitoring of the Training Load and Well-Being of Elite Rhythmic Gymnastics Athletes in 25 Weeks: A Comparison between Starters and Reserves"

_sports, 2022, doi:10.3390/sports10120192_

Round 1

Reviewer 1 Report

Sports-1841778

Monitoring of the training load and well-being of high performance rhythmic gymnastics athletes in different periods: comparison between starters and reserves

This study examined differences in the training load, and well-being between starters and reserves over three training periods (basic and specific preparatory, and pre-competitive).

Although the main concept of the study is interesting there are several issues that should be addressed:

1.       the methods section is poorly written, and it is difficult for the reader to understand the tests used and the result of the study.

Please, consider rewriting this section.

2.        Although, monitoring that training load of elite athletes is very interesting, the fact that 5 starters and 5 reserves took part in the study is a limitation that should be mentioned.

3.       I feel that the authors should provide a better rationale for the study.

4.       A language revision throughout the study would be beneficial

ABSTRACT

1.       Please report briefly how TL, duration and monotony were assessed

2.       Please include in the abstract numerical results or/and exact p values (e.g…not p<0.05) for the examined parameters, so that the reader can understand the magnitude of the differences between starters and reserves.

INTRODUCTION

3.       In general: please provide a better rationale for the study.

The aim of the study is interesting, but I feel that the connection to the existing literature is missing.

METHODS

1.       Please, provide maturity offset for the participants.

2.       Please provide details for each test or questionnaire used.

In particular: how was the TL measured? Strain? Monotony? Please, also provide for all the examined variables reliability values.

3.       Please provide a schematic representation of the study protocol. It not clear when and which weeks were assessed.

4.       How was well being measured? Please provide the reference for the questionnaire used.

Reviewer 2 Report

The manuscript is clear and relevant for the field and it is well structured and organized. The cited references are of latest publications in the topic.

It is scientifically correct, proper and clear descriptive study design, data collection and variables description.

Results are consistent with the aim of the study and well presented. The table and figures are easy to understand by the reader and well summarize the main findings. The conclusions are consitent with the evidence and the arguments presented. 

The ethics statement is adequate while data avalability is under ethical restrictions.

Overall is a good descriptive study and article that contributes to a better knowledge of the scientific basis for using training loads protocols in high level Rhythmic Gymnastics, complex sport difficult to quantify and measure.

Author Response

Dear reviewer, we appreciate your careful review of our manuscript. 

Reviewer 3 Report

Dear Authors,

You conducted interesting research. The manuscript consists some unclarity which should be explain.

Comments and questions:

1.     Key words: Why did you use the words: ʺsubjectiveʺ, ʺteam sportsʺ? These words do not seem appropriate for this manuscript. Please consider adding adding the word ʺrhythmic gymnasticsʺ.

2.     Introduction: The line ʺthe code of pointsʺ. It was repeated in this same line. This is a stylistic error.

3.     Line 55 ʺAUʺ – it should be explain in this place (not only in the table).

Materials and Methods

4.     Subject: you describe only one group (starters). Why did not describe the group „reserves”? It should be clearly stated whether there was one or two groups.

5.     Line 81 – it should be added: hight and body mass.

6.     Statistical analysis: please add „range”.

Results

7.     Figure 1 A-E, Figure 2, Figure 3 A-E are too small. They are illegible. They are more ʺillustrativeʺ than showing the results. Consider improving tables and figures.

8.     Discussion: The first paragraph is more summary of findings than discussion.

9.     Conclusion should only refer to conducted research. Any other considerations should be posted in the chapter Discussion. Examine line 337-345 again.

Round 2

Reviewer 1 Report

I have no further comments